# Dose-Dependent Effect of Cordycepin on Viability, Proliferation, Cell Cycle, and Migration in Dental Pulp Stem Cells

**DOI:** 10.3390/jpm11080718

**Published:** 2021-07-26

**Authors:** Nezar Boreak, Ahmed Alkahtani, Khalid Alzahrani, Amani Hassan Kenani, Wafa Hussain Faqehi, Hadeel Hussain Faqehi, Raghad Essa Ageeli, Wafa Nasser Moafa, Hosam Ali Baeshen, Shilpa Bhandi, Zohaib Khurshid, Vikrant R. Patil, Luca Testarelli, Shankargouda Patil

**Affiliations:** 1Department of Restorative Dental Sciences, College of Dentistry, Restorative Dental Sciences Jazan University, Jazan 45142, Saudi Arabia; nezarboreak@gmail.com (N.B.); shilpa.bhandi@gmail.com (S.B.); 2Department of Restorative Dental Sciences, College of Dentistry, King Saud University, Riyadh 11362, Saudi Arabia; ahkahtani@ksu.edu.sa; 3Department of Clinical Laboratories Sciences, College of Applied Medical Sciences, Taif University, Taif 21944, Saudi Arabia; ak.jamaan@tu.edu.sa; 4College of Dentistry, Jazan University, Jazan 45142, Saudi Arabia; Amanihassankenani@gmail.com (A.H.K.); Wafaafaqehi@gmail.com (W.H.F.); hadeel937@iCloud.com (H.H.F.); raghadageeli3@gmail.com (R.E.A.); wafa0954@gmail.com (W.N.M.); 5Department of Orthodontics, College of Dentistry, King Abdulaziz University, Jeddah 11451, Saudi Arabia; drbaeshen@me.com; 6Department of Prosthodontics and Dental Implantology, College of Dentistry, King Faisal University, Al-Ahsa 31982, Saudi Arabia; drzohaibkhurshid@gmail.com; 7Biogenre Private Limited, Pune 412105, India; vikrant@biogenre.com; 8Department of Oral and Maxillofacial Sciences, Sapienza University of Rome, 00185 Rome, Italy; luca.testarelli@uniroma1.it; 9Division of Oral Pathology, Department of Maxillofacial Surgery and Diagnostic Sciences, College of Dentistry Jazan University, Jazan 45142, Saudi Arabia

**Keywords:** cell cycle, Cordycepin, dental pulp stem cells, growth curve

## Abstract

**Simple Summary:**

Cordycepin is an adenosine analogue isolated from the fungus *Cordyceps militaris*. Cordycepin is a nucleoside antimetabolite that has shown a broad spectrum of biological activity including antineoplastic activity. limited research has been carried out on the effects of Cordycepin on the regenerative potential of stem cells, including dental pulp-derived mesenchymal stem cells. The present study was designed to assess if Cordycepin could enhance the vital properties of dental pulp-derived mesenchymal stem cells for regenerative purposes.

**Abstract:**

Objective: To examine the effect of Cordycepin on the viability, proliferation, and migratory properties of dental pulp-derived mesenchymal stem cells. Materials and methods: The pulp was derived from human premolar teeth extracted for orthodontic purposes after obtaining informed consent. The samples were transferred to the laboratory for processing. DPSCs were expanded and characterized using flow cytometry and differentiation to the bone, adipose, and cartilage cells was examined. MTT Assay was performed using various concentrations of Cordycepin. The growth curve was plotted for 13 days. Cell cycle analysis was performed by flow cytometry. Migratory ability was assessed by wound healing assay. ROS generation was detected by flow cytometry. Gene expression was quantified by RT-qPCR. Statistical analysis was performed. *p* < 0.05 was considered as significant and *p* < 0.01 was considered as highly significant (* *p* < 0.05, and ** *p* < 0.01). Results: DPSCs expressed characteristic MSC-specific markers and trilineage differentiation. Cordycepin at lower concentrations did not affect the viability of DPSCs. The growth curve of cells showed a dose-dependent increase in cell numbers till the maximum dose. DPSCs treated with 2.5 µM Cordycepin was found to have a reduced G1 phase cell percentage. DPSCs treated with 2.5 µM and 5 µM Cordycepin showed a significant decrease in G2 phase cells. No significant difference was observed for S phase cells. Cordycepin treatment affected the migratory ability in DPSCs in a concentration-dependent manner. Conclusion: Cordycepin can be used at therapeutic doses to maintain stem cells.

## 1. Introduction

Biometabolites derived from plants, animals, fungi, and algae have been used in medicine throughout history. Plant and animal extracts have found use as preventive and therapeutic agents. In the modern era, they have been important primary lead molecules for the development of potent drugs. The therapeutic effects of medicinal mushrooms have been recorded by various cultures, and have formed the basis for the synthesis of a large number of pharmaceutical products [1].

Cordycepin (Cpn), (3′-deoxyadenosine) is derived from *Cordyceps militaris*, a medicinal mushroom. Cordycepin has a broad spectrum of biological actions, including antineoplastic activity [2]. It resembles the nucleoside adenosine and is shown to kill neoplastic cells by interfering with their metabolism. Cordycepin has effects on the cardiovascular, respiratory, nervous, sexual, and immunological systems, and shows anti-oxidant, anti-inflammatory, and anti-microbial activities [3,4].

Multipotent dental pulp stem cells (DPSC) can be derived from extracted teeth. There has been a recent surge of interest in DPSCs due to their potential applications in dental tissue engineering and regenerative medicine. Their ecto-mesenchymal origin allows these cells to differentiate into various lineages such as bone, nerve, and cartilage [5]. Various bioactive components are needed to maintain, migrate, and differentiate these multipotent cells. There is evidence that these components can influence the turnover and migration of the stem cells. 

Recent developments in the application of DPSCs in various therapeutics have highlighted the need for further research. There is limited published literature on the effect of Cordycepin on the viability, proliferation, and migratory properties of dental pulp-derived mesenchymal stem cells [2]. Data about the impact and efficacy of Cordycepin on DPSCs can guide future research and unearth new therapeutic applications for the drug. The present study examined the effects of Cordycepin on the viability, proliferation, and migratory characteristics of dental pulp stem cells.

## 2. Materials and Methods

### 2.1. Sample Collection

Informed consent was obtained from all subjects following institutional ethics considerations before extraction of teeth (Scientific research, College of Dentistry, Jazan University (Reference no:19710)). Human premolar teeth were obtained from healthy subjects who were aged 14–25 years with good oral hygiene undergoing orthodontic tooth extraction. The pulp was extracted in sterilized conditions and transferred to the laboratory directly for further processing.

### 2.2. Culture and Expansion of Human DPSCs

Isolation and characterization of DPSCs were carried out using the explant culture method [6]. Pulp tissue was minced into tiny fragments. The pieces were placed in 35 mm polystyrene plastic culture dishes. A sufficient amount of fetal Bovine Serum (FBS) (Gibco, Rockville, MD, USA) was added to the tissues to cover them completely. The tissues were incubated for 24 h at 37 °C and 5% CO_2_. The DPSCs culture system was maintained in DMEM (Invitrogen, Carlsbad, CA, USA) supplemented with 20% FBS and antibiotic-antimycotic solution at the same temperature and CO_2_ conditions. The culture medium was replenished twice weekly, and the cell growth, health, and morphology were evaluated regularly with an inverted phase-contrast microscope. At 70–80% confluence, the cells were detached using 0.25% Trypsin-EDTA solution (Invitrogen, Carlsbad, CA, USA) and transferred to a 25-cm^2^ polystyrene culture flask (Nunc, Rochester, NY, USA). Confluent DPSCs were detached using 0.25% Trypsin-EDTA solution, and then continuously passaged in for expansion and further experiments. Cells from passages 2 to 4 were used in the experimentation.

### 2.3. Characterization of DPSCs by Flow Cytometry

For cell surface marker analysis, DPSCs (Confluent) were collected by trypsinization and washed twice with Phosphate Buffered Saline. The cells were then incubated at 4 °C for 30 min with anti-human-CD73-APC, anti-human-CD90-APC, anti-human-CD105-APC, anti-human-CD34-PE, anti-human-CD45-FITC, and anti-human-HLA-DR-APC antibodies (Miltenyi Biotec, Bergisch Gladbach, Germany). Antibody-stained cells were washed twice with PBS. A total of 10,000 cells per sample were acquired on Attune NxT Flow Cytometer (Thermo Fisher Scientific, Waltham, MA, USA). Isotype controls were used for the detection of and to differentiate between positive and negative signals.

### 2.4. Osteogenic Differentiation

The cells were seeded in a 24-well plate (Nunc, Rochester, NY, USA) at a density of 2500 cells/cm^2^. The cells were initially seeded with the complete growth medium. After 24 h, the medium was replaced with osteogenic induction medium that consisted of DMEM with 1% antibiotic-antimycotic, 50 µM of ascorbate-2-phosphate, 0.1 µM of dexamethasone, and 10 mM of β-glycerophosphate (Sigma-Aldrich Corp., St. Louis, MO, USA). A fresh induction medium of similar composition was used to replace the older media twice a week. After 21 days, osteogenic differentiation was assessed by fixing the cells using paraformaldehyde (4%), and staining with 2% alizarin red S (pH 4.1–4.3) for 20 min. 

### 2.5. Adipogenic Differentiation

The cells were seeded in a 24-well plate (2500/cm^2^) (Nunc, Rochester, NY, USA) with a complete growth medium. After 24 h, the complete growth medium was replaced with adipogenic media, DMEM supplemented with 10% FBS, 10 µM insulin, 1 µM dexamethasone, 200 µM indomethacin, and 0.5 mM isobutyl-methylxanthine (Sigma-Aldrich Corp., St. Louis, MO, USA). The medium was introduced to the cells twice a week for three weeks. Three experimental groups were created: control, induction, and 5 µM Cordycepin treatment with induction. Differentiated adipocytes were fixed with 4% paraformaldehyde, and confirmed using 0.3% oil red O for oil droplets for 1 h.

### 2.6. Chondrogenic Differentiation

The cells were seeded in a 24-well plate (Nunc, Rochester, NY, USA) at a density of 2500 cells/cm^2^ with the complete growth medium. After 24 h, the complete growth medium was replaced with a chondrogenic induction medium, DMEM with 1X-ITS, 1 mM of sodium pyruvate, 100 nM of dexamethasone, 50 µg/mL of ascorbate-2-phosphate, 40 µg/mL of L-proline, and 10 ng/mL of TGF-β3 (Sigma-Aldrich Corp., St. Louis, MO, USA). Cultures were incubated for 28 days at 37 °C at 5% CO_2_. The culture medium was replaced every 2–3 days. To analyze differentiation towards chondrogenic lineage, alcian blue staining was performed on fixed cells after 28 days. Cells were then fixed with 4% paraformaldehyde and stained with 0.1% alcian blue for 30 min.

### 2.7. MTT Assay of DPSC following Treatment

The cells were seeded in 96-well plates (1 × 10^4^ cells per well) and subjected to the treatment with various concentrations of Cordycepin (Cpn) (Sigma Aldrich, St. Louis, MO, USA) (0.5 µM, 1 µM, 2.5 µM, 5 µM, 10 µM, 25 µM, and 50 µM). The cytotoxicity of Cordycepin to DPSCs was measured using the MTT assay. The cells were seeded into 96-well plates and incubated with appropriate media for 24, 48, and 72 h. Following incubation, MTT solution (Sigma-Aldrich Corp., St. Louis, MO, USA) at a concentration of 0.5 mg/mL was mixed in each well. The mixing plates were incubated for 4 h at 37 °C. After incubation, the medium was removed, and 100 µL dimethyl sulfoxide (DMSO) (Sigma-Aldrich Corp., St. Louis, MO, USA) was added to each well. The absorbance was measured at 570 nm using a Multiskan Spectrum spectrophotometer (Thermo Scientific, San Jose, CA, USA).

### 2.8. Growth Curve Plotting

To check the proliferative aptitude of DPSCs, 1 × 10^4^ cells at passage 2 were seeded into 12-well cell culture plates. The cell count was estimated each day for a total of 13 days. The growth curve was plotted using cell numbers counted for 13 days.

### 2.9. Cell Cycle Analysis by Flow Cytometry

Assessment of cell cycle phase distribution by DNA content was carried out by using flow cytometry. The DPSCs were plated in 12-well plates at a density of 5 × 10^4^ cells per well and incubated for 24 h. The cells were treated with Cpn for 48 h. Cultured cells were subsequently washed with PBS, fixed in 70% ethanol at 20 °C for 2 h. The washed cells were then treated with RNase A (10 mg/mL) and stained with propidium iodide (PI) in the dark for 30 min at room temperature. PI fluorescence of individual nuclei was measured using flow cytometry and the percentage of the cells in different cell cycle phases was calculated.

### 2.10. Analysis of Migration by Wound Scratch Assay

For the measurement of cell migration, confluent DPSCs were incubated at 37 °C and 5% CO_2_ in a serum-free medium for 24 h and mitotically inactivated with 10 µg/mL of mitomycin C (Sigma Aldrich, St. Louis, MO, USA) for 2 h. Wound scratches were created with a sterile plastic 200 μL micropipette tip. After washing, the medium was replaced with a fresh complete growth medium. Photographs of the wound area were taken at 0 h and 24 h under a microscope. The borders along each wound were marked to evaluate wound closure. The horizontal distance of migrating cells from the initial wound was measured. The percentage was calculated from the distances measured.

### 2.11. ROS Assay

The intracellular reactive oxygen species generated during prolonged incubation for 7 days was measured using a Cellular ROS/Superoxide detection assay kit (Abcam, Cambridge, MA, USA). Then, 1 × 10^4^ cells were seeded into 12-well cell culture plates and incubated for 24 h. The cells were treated with Cpn. The media and Cpn were replaced every alternate day. After 7 days of incubation, the cells were incubated with a 2 μM ROS/Superoxide detection mix for 30 min at 37 °C in darkness. The cells were washed twice with washing buffer and immediately acquired on a flow cytometer. 

### 2.12. Real-Time Quantitative PCR for Analysis of Gene Expression

The total RNA was extracted from the cells by using the GeneJET RNA purification kit (Thermo Scientific, Lithuania). RNA (2 μg) was reverse transcribed using a cDNA synthesis kit (High Capacity, Applied Biosystems, Carlsbad, CA, USA) according to the manufacturer’s guidelines. Total 100 ng cDNA was used for the total reaction volume of 20 μg for each gene. Quantitative analysis of genes of interest was carried out using the SYBR Green PCR master mix (Applied Biosystems, Austin, TX, USA) on a Real-Time PCR system (QS5, Applied Biosystems, Foster City, CA, USA). Expressions of target genes related to stemness, pluripotency, and differentiation were normalized to GAPDH as a reference gene using the ΔΔCt method. Genes and primers (IDT, Coralville, IA, USA) are listed in Table 1.

### 2.13. Statistical Analysis

The results were presented as the mean ± standard deviation of the values from two independent experimental values. Each treatment group was individually compared with the control group. The data were analyzed using an unpaired t-test (two-tailed) on GraphPad Prism 8 software (GraphPad Software, La Jolla, CA, USA); *p* < 0.05 was considered as significant and *p* < 0.01 was considered as highly significant (ns not significant, * *p* < 0.05, and ** *p* < 0.01).

## 3. Results

### 3.1. DPSCs Show Characteristic Marker Expression for MSC-Specific Markers and Trilineage Differentiation

DPSCs showed MSC-like morphology (Figure 1A) and positive expression for MSC-specific markers. DPSCs showed a positive expression of CD73, CD90, and CD105 (Figure 1B–E). Non-MSC markers such as CD34, CD45, and HLA-DR showed negative expression in DPSCs (Figure 1F–H). DPSCs differentiated into adipocytes, osteoblasts, and chondrocytes, validating the successful isolation of DPSCs (Figure 1I–K).

### 3.2. Cordycepin at Lower Concentrations Does Not Affect the Viability of DPSCs

DPSCs were treated with different concentrations of Cordycepin (0.5 µM, 1 µM, 2.5 µM, 5 µM, 10 µM, 25 µM, and 50 µM). Cell viability was assessed using MTT assay after 24, 48, and 72 h of treatment. Lower concentrations of Cordycepin at 0.5 µM, 1 µM, 2.5 µM, and 5 µM showed no significant difference in the viability of DPSCs. However, higher concentrations of Cordycepin at 10 µM, 25 µM, and 50 µM showed a cytotoxic effect on DPSCs (Figure 2A–C). Amounts of 0.5, 1, 2.5, and 5 µM of Cordycepin were used for all the experiments.

### 3.3. Growth Curve

The growth curve obtained by calculating the number of cells from day 1 to day 13 showed that 1 µM Cpn increased the cell number significantly at day 5, whereas 2.5 µM and 5 µM Cpn showed a significant increase in the cell number at day 7 (Figure 2D–G).

### 3.4. Cell Cycle Analysis by PI Staining in DPSCs Treated with Various Concentrations of Cpn 

Cell cycle analysis was achieved by flow cytometry. The 2.5 µM Cpn treated DPSCs were found to be decreased for G1 phase (Figure 3A–G) cell percentage, and 2.5 µM and 5 µM Cpn treated DPSCs showed a significant decrease, observed for G2 phase cells (Figure 3A–F,I). No significant difference was observed for S phase cells (Figure 3A–F,H).

### 3.5. Cordycepin Treatment Affects the Migratory Ability in DPSCs in a Concentration-Dependent Manner

A wound-healing assay was performed to assess the migratory ability of DPSCs treated with different concentrations of Cpn. It was observed that Cpn at a lower concentration of 0.5 µM showed significantly decreased migration, whereas Cpn at a higher concentration of 5 µM showed significantly increased migration. (Figure 4A–K)

### 3.6. Cordycepin Treatment Reduces Reactive Oxygen Species Generation in DPSCs at Higher Concentration

Total cellular ROS was detected in DPSCs treated with different concentrations of Cpn for a longer period of 7 days. It was observed that Cpn at a higher concentration of 5 µM showed significantly decreased ROS percentage (Figure 5A–G).

### 3.7. Cordycepin Treatment for a Longer Period of 7 Days Changes the Morphology of DPSCs, Downregulates Gene Expression of Adipogenesis and Chondrogenesis Master Regulator Genes, Upregulates Gene Expression of Osteogenesis Master Regulator, and Shows Differential Effects on the Gene Expression of Stemness Related Transcription Factors

DPSCs were treated with different concentrations of Cpn and incubated for 7 days. It was observed that Cpn showed significantly changed morphological differences at different concentrations (Figure 6A–E). Cpn at a higher concentration of 5 µM showed significantly decreased PPARG expression, but increased RUNX2 expression (Figure 6F–H). Cpn at lower concentrations showed downregulated SOX9 expression. However, Cpn showed differential changes in stemness-related transcription factors SOX2, NANOG, and OCT4 in a concentration-dependent manner (Figure 6I–K).

## 4. Discussion

Previous studies have focused on multifaceted effects of Cordycepin on cancer cells. Yoon et al., [4] reviewed the effects and mechanisms of Cordycepin. They reported the induction of apoptosis and arrest of the cell cycle. Cordycepin is a nucleoside-analogue with the ability to interact with DNA and/or RNA polymerases. After entering a cell, it is converted to nucleosides such as mono, di, or tri-phosphates of 3-deoxyadenosine. As there is a resemblance of structure with adenosine monophosphate, Cordycepin monophosphate may cause the termination of elongation of nucleic acid chains. This is demonstrated in yeast and mammals [7,8]. 

Cordycepin may have applications in bone regeneration [9]. Yu et al. [10] reported on the dual role of Cordycepin. It promoted anabolic changes in bone via BMP2/Runx2/Osterix pathway. At doses of 1 μM concentration, Cpn significantly inhibited RANKL-induced osteoclast formation. Wang et al. [11] observed that Cordycepin can prevent inhibition of osteogenesis induced by oxidative stress. Yang et al. [12] examined the effects of Cordycepin on human adipose-derived stem cells and suggested that Cordycepin had protective action against the osteogenic inhibition induced by TNF-α at 10 μg/mL concentrations. 

It is now well established that Cordycepin affects stem cells. The anabolic effects of the drug on bone tissue make it of interest to dentists and periodontists dealing with periodontal and periapical bone regeneration. In this context, therapeutic doses of Cordycepin may help regenerate these tissues [13]. The data from this study may help determine the optimum dosage of Cordycepin for stem cell maintenance and migration. 

First, DPSCs were isolated and confirmed using the positive expression of CD73, CD90, and CD105, along with negative expression of non-MSC markers CD34, CD45, and HLA-DR. Lower concentrations of Cordycepin up to 5 µM did not affect the viability of DPSCs. However, concentrations above 10 µM showed the cytotoxic effect. Wang et al. reported similar results in embryonic stem cells [14]. 

In the present study, 1 µM Cpn was seen to increase the cell number significantly at day 5. Concentrations of Cordycepin at 2.5 µM and 5 µM resulted in a significant increase in the number of cells at day 7. Flow cytometry analysis showed that 2.5 µM Cpn treated DPSCs were decreased for G1 phase cell percentage. Cells that were treated with 2.5 µM and 5 µM Cpn showed a significant decrease for G2 phase cells. No significant difference was observed for S phase cells. 

Migratory ability was assessed using a wound-healing assay. Results revealed a dose-dependent increase in migration, which was not linearly related to dose (Figure 4). Previous authors have reported a reduction in migration of cancer cells in the presence of Cordycepin [15,16]. However, our study reports an increase in migration of stem cells. The results of our study suggest that the optimum dosage of Cpn can be used to maintain DPSC. However, it seems uncertain whether these results can be generalized for all types of stem cells.

Despite these promising results, questions remain on the effect of Cordycepin on various stem cells. Further research exploring the mechanisms of action of Cordycepin will shed light on its effects on different stem cells. Recently, phytochemicals have been examined for application in regeneration [17]. Many plant-derived metabolites have been investigated as an alternative to bone morphogenetic proteins [18]. The use of Cpn shows therapeutic potential in regenerative medicine. A limitation of the current study is that it does not elucidate the mechanism of action of the effects of Cordycepin. 

## 5. Conclusions

This study set out to establish the effects of various concentrations of Cordycepin on DPSCs. DPSCs could survive Cpn at concentrations below 10 uM. The growth curve of the cells showed a dose-dependent increase until the maximum dose. DPSCs treated with lower concentrations of Cordycepin resulted in a reduced G1 phase cell percentage. DPSCs treated with 2.5–5 µM concentrations Cpn showed a reduction in the G2 phase. Cordycepin treatment showed no significant difference to cells in the S phase. These findings suggest that Cordycepin affects the migratory ability in DPSCs in a concentration-dependent manner.

## Figures and Tables

**Figure 1 jpm-11-00718-f001:**
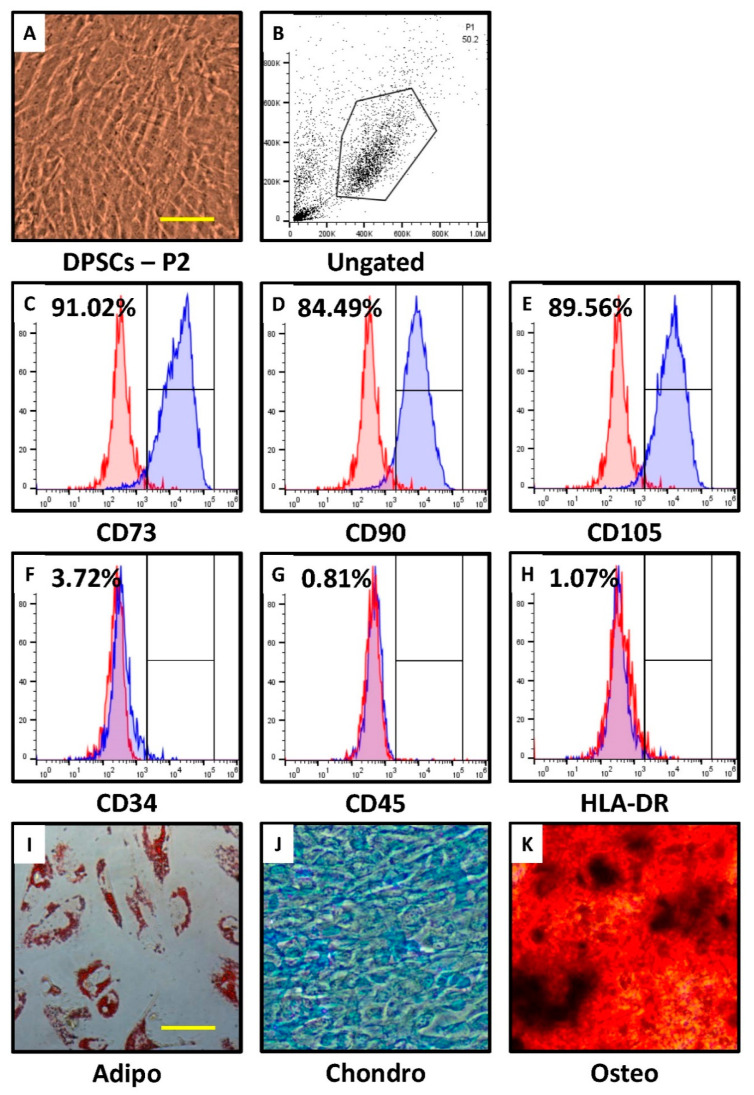
Characterization of DPSCs for mesenchymal stem cell properties. (**A**) Photomicrograph of DPSCs at passage 2. Scale bar = 100 μm, (**B**–**H**) DPSCs were checked for MSC-specific positive markers CD73, CD90, and CD105; and MSC-specific negative markers CD34, CD45, and HLA-DR. (**I**–**K**) Differentiation of GMSCs into adipocytes, osteoblasts, and chondrocytes. Scale bar = 100 µm.

**Figure 2 jpm-11-00718-f002:**
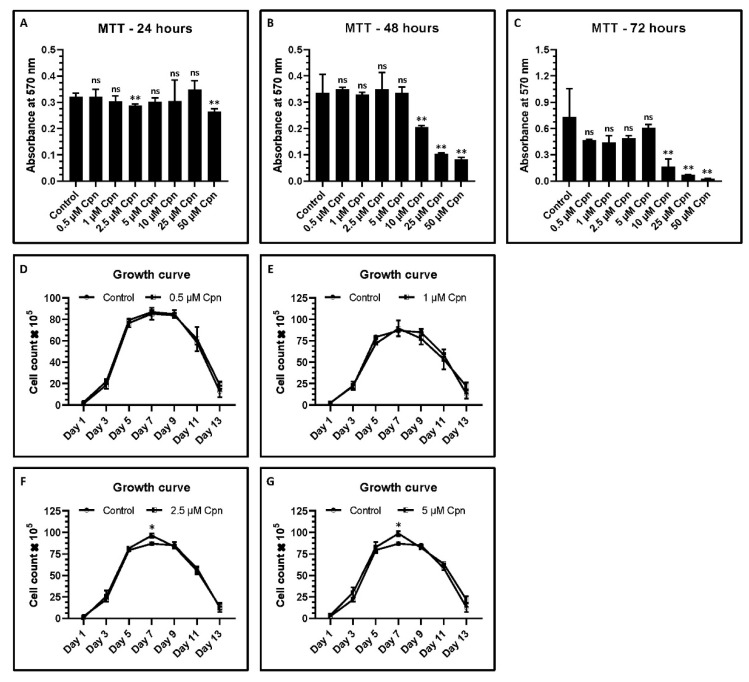
MTT assay and growth curve plotting. (**A**–**C**) DPSCs were treated with various concentrations of Cordycepin (0.5 μM, 1 μM, 2.5 μM, 5 μM, 10 μM, 25 μM, and 50 μM) for 24, 48, and 72 h and comparative analysis was performed to check the cytotoxicity of Cordycepin to DPSCs. (**D**–**G**) Growth curves of DPSCs treated with various concentrations of cordycepin (0.5 μM, 1 μM, 2.5 μM, and 5 μM). ns not significant, * *p* < 0.05, and ** *p* < 0.01. Cpn: Cordycepin.

**Figure 3 jpm-11-00718-f003:**
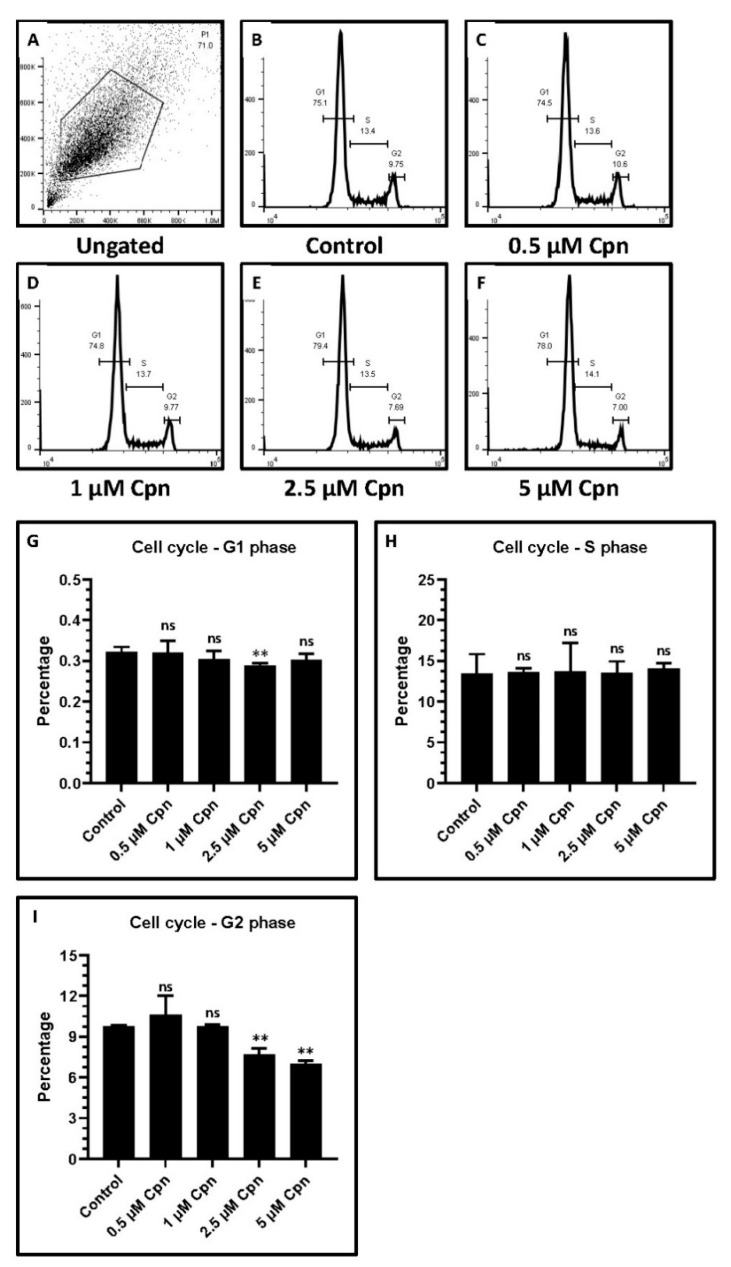
Cell cycle analysis of DPSCs treated with various concentrations of Cordycepin (0.5 μM, 1 μM, 2.5 μM, and 5 μM). (**A**–**F**) Cell cycle analysis. (**G**) Comparative analysis of G1 phase cells. (**H**) Comparative analysis of G2 phase cells. (**I**) Comparative analysis of S phase cells. ns not significant, ** *p* < 0.01. Cpn: Cordycepin.

**Figure 4 jpm-11-00718-f004:**
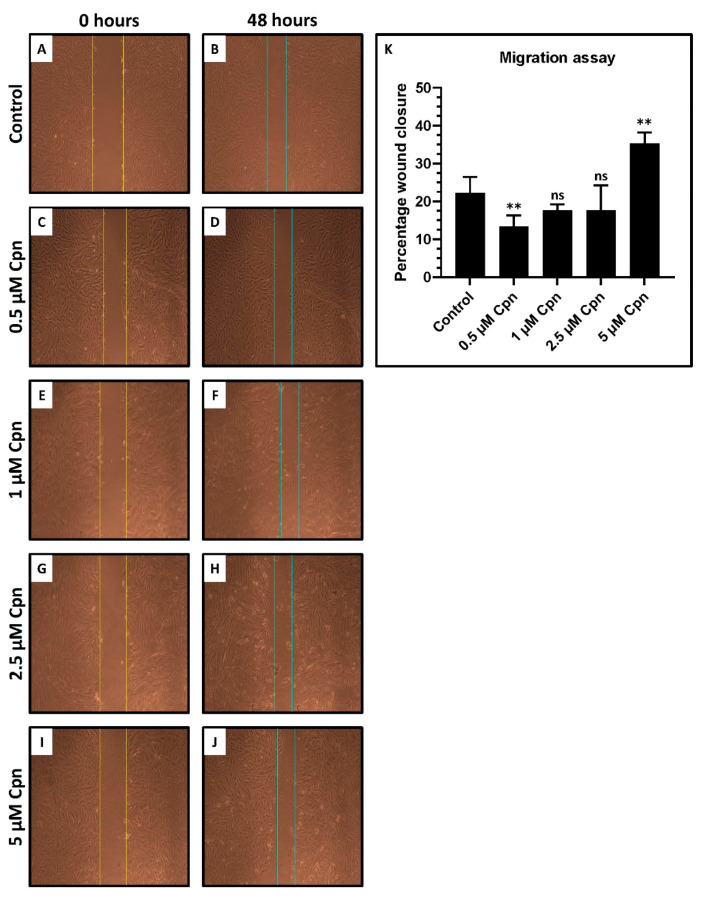
Wound healing scratch assay to assess the migratory ability of DPSCs treated with various concentrations of cordycepin (0.5 μM, 1 μM, 2.5 μM, and 5 μM). (**A**–**J**) Scratch assay. (**K**) Comparative analysis of percentage migration of DPSCs. ns not significant, ** *p* < 0.01. Cpn: Cordycepin.

**Figure 5 jpm-11-00718-f005:**
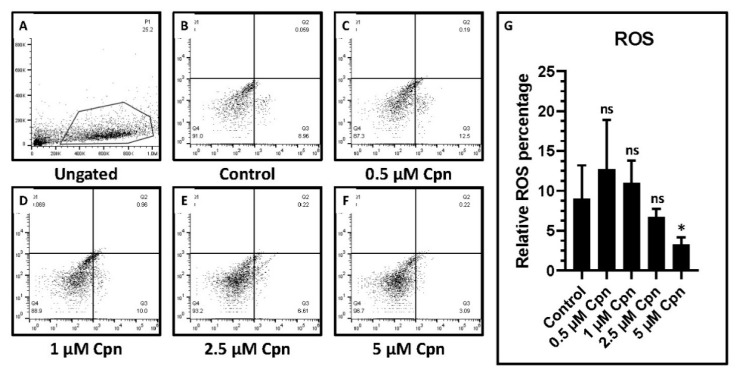
Analysis of ROS in DPSCs treated with various concentrations of Cordycepin (0.5 μM, 1 μM, 2.5 μM, and 5 μM) for a prolonged time of 7 days. (**A**–**F**) ROS analysis. (**G**) Comparative analysis of relative ROS percentage. ns not significant, * *p* < 0.05, Cpn: Cordycepin, ROS: Reactive oxygen species.

**Figure 6 jpm-11-00718-f006:**
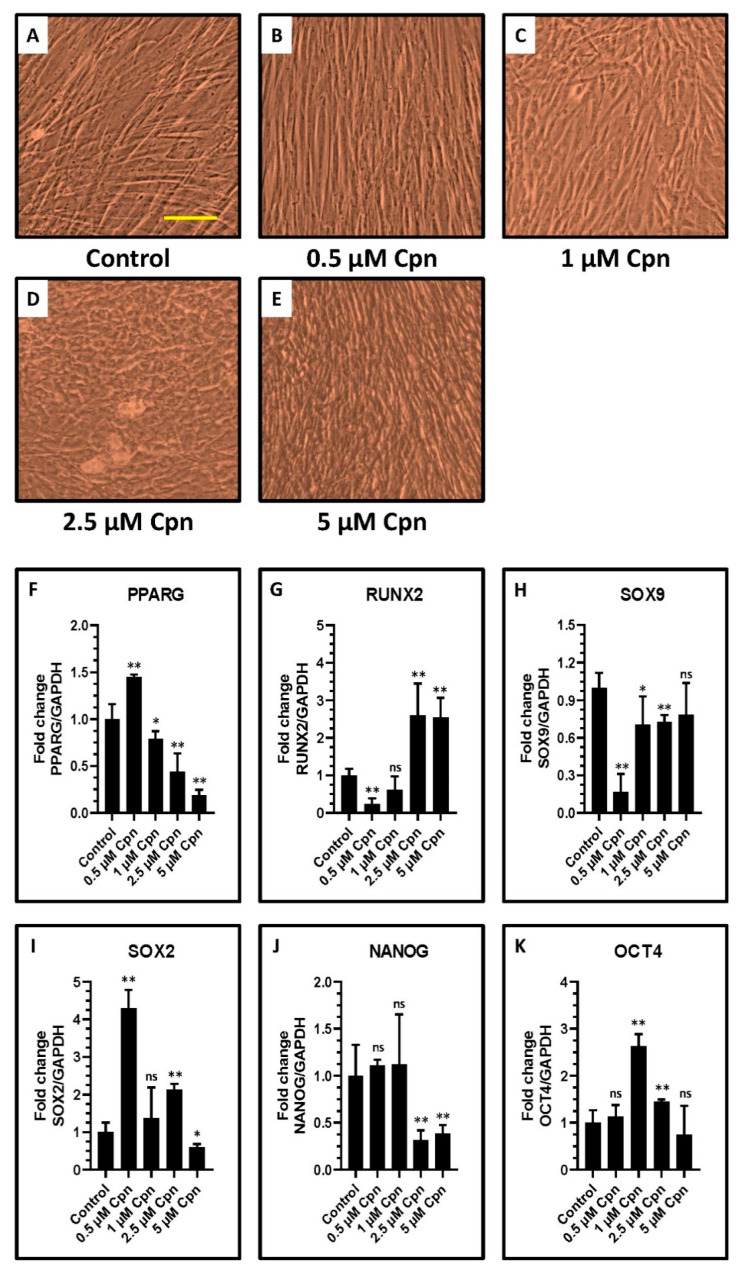
Morphology and analysis of gene expression in DPSCs treated with various concentrations of cordycepin (0.5 μM, 1 μM, 2.5 μM, and 5 μM) for a prolonged time of 7 days. (**A**–**E**) Photomicrograph of DPSCs after 7 days of incubation with various treatments. (**F**–**H**) Comparative gene expression analysis of PPARG (adipogenesis master regulator gene), RUNX2 (osteogenesis master regulator gene), SOX9 (chondrogenesis master regulator gene). (**I**–**K**) Comparative gene expression analysis of SOX2, NANOG, and OCT4 (stemness-related transcription factors). ns not significant, * *p* < 0.05, and ** *p* < 0.01. Cpn: Cordycepin, ROS: Reactive oxygen species, PPARG: Peroxisome proliferator-activated receptor-gamma, RUNX2: Runt-related transcription factor 2, SOX9: SRY-box transcription factor 9, SOX2: SRY-box transcription factor 2, NANOG: Nanog homeobox, and OCT4: Octamer-binding transcription factor 4.

**Table 1 jpm-11-00718-t001:** List of primers.

Gene	Forward Primer	Reverse Primer
PPARG	5′-AGC CTG CGA AAG CCT TTT GGT G-3′	5′-GGC TTC ACA TTC AGC AAA CCT GG-3′
SOX9	5′-GCC GAA AGC GGG CTC GAA AC-3′	5′-AAA AGT GGG GGC GCT TGC ACC-3′
RUNX2	5′-GTG CCT AGG CGC ATT TCA-3′	5′-GCT CTT CTT ACT GAG AGT GGA AGG-3′
SOX2	5′-GCT ACA GCA TGA TGC AGG ACC A-3′	5′-TCT GCG AGC TGG TCA TGG AGT T-3′
NANOG	5′-CTC CAA CAT CCT GAA CCT CAG C-3′	5′-CGT CAC ACC ATT GCT ATT CTT CG-3′
OCT4	5′-CCT GAA GCA GAA GAG GAT CAC C-3′	5′-AAA GCG GCA GAT GGT CGT TTG G-3′
GAPDH	5′-GTC TCC TCT GAC TTC AAC AGC G-3′	5′-ACC ACC CTG TTG CTG TAG CCA A-3′

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
