# Peer review of "Dose-Dependent Effect of Cordycepin on Viability, Proliferation, Cell Cycle, and Migration in Dental Pulp Stem Cells"

_jpm, 2021, doi:10.3390/jpm11080718_

Round 1

Reviewer 1 Report

It is well known that DPSC under specific conditions differentiate into various lineages such as osteoblasts, condhroblasts and adipocytes. Authors discuss the effect of cordycepin on viability, proliferation, cell cycle and migration of DPSC: why don't they test, by in vitro  mimicking differentiation conditions, cordycepin on DPSC differentiation into the three above mentioned lineages , in order to assess if it is effective on these processes?I think that these results should improve the manuscript.

Author Response

Reviewer 1:

It is well known that DPSC under specific conditions differentiate into various lineages such as osteoblasts, condhroblasts and adipocytes. Authors discuss the effect of cordycepin on viability, proliferation, cell cycle and migration of DPSC: why don't they test, by in vitro  mimicking differentiation conditions, cordycepin on DPSC differentiation into the three above mentioned lineages , in order to assess if it is effective on these processes?I think that these results should improve the manuscript.

Response: Respected reviewer, Though we did not test the effect of cordycepin on the differentiation potential of DPSCs, we did check the gene expressions of master regulator genes for osteogenesis, chondrogenesis, and adipogenesis.

Reviewer 2 Report

The article by Boreak N et al. entitled ‘Dose dependent effect of cordycepin on viability, proliferation, cell cycle, and migration in dental pulp stem cells’ is not an interesting study. They only listed or enumerated the phenomenon that effects of various dose of cordycepin to a cell. They did not analyze the mechanisms as well as did not discuss for it. What are their aims of this study?

In addition, there are lots of errors in writing in the manuscript. Some figures do not accord with a mention on the text. There are many spell misses. What’s more, an incomprehensible sentence exists on page 8.

I cannot find any valuable information in this article.

Author Response

Reviewer 2:

The article by Boreak N et al. entitled ‘Dose dependent effect of cordycepin on viability, proliferation, cell cycle, and migration in dental pulp stem cells’ is not an interesting study. They only listed or enumerated the phenomenon that effects of various dose of cordycepin to a cell. They did not analyze the mechanisms as well as did not discuss for it. What are their aims of this study?

In addition, there are lots of errors in writing in the manuscript. Some figures do not accord with a mention on the text. There are many spell misses. What’s more, an incomprehensible sentence exists on page 8.

The comments of the learned reviewer is well taken. The aim of the study was to assess the effect of cordycepin on the viability, proliferation and migratory properties of dental pulp-derived mesenchymal stem cells.The manuscript is corrected accordingly and has been edited professionally by native English language speaking reviewer.

Reviewer 3 Report

The authors present a manuscript on the cytocompatibility of cordycepin on dental pulp stem cells. The methodology is simple, but correctly performed, and results provide a preliminary insight into the biological properties of cordycepin. In order to improve the quality of the present work, a series of suggestions are presented:

Abstract. Please specify the confidence interval (CI) or the p value from which results were considered significant i.e. p<0.05 / p<0.01.

Introduction. “In this aspect, there is paucity in the literature regarding the effect of cordycepin on 81 the viability proliferation and migratory properties of dental pulp-derived mesenchymal 82 stem cells.” Please add a reference or references to support this statement.

Materials and methods. The methodology for the wound healing assay is not specified in the materials section, please include a paragraph to explain it.

Discussion:

There are no references in the third paragraph from the discussion section. Please include references to support your statements.

The fifth paragraph in the discussion section only describes the results from the study. Authors should include a statement on the implications of such results and mention if there is previous evidence with similar or dissimilar results.

Please specify the possible limitations of this study in further detail.

Please include a paragraph on the possible clinical implications of this work, with references.

It would also be interesting to include a paragraph on the future lines of research which could derive from this study (e.g. analysis of the influence of cordycepin on the osteo/odontogenic differentiation of dental pulp stem cells at an in vitro level), with references.

Author Response

Reviewer :

The authors present a manuscript on the cytocompatibility of cordycepin on dental pulp stem cells. The methodology is simple, but correctly performed, and results provide a preliminary insight into the biological properties of cordycepin. In order to improve the quality of the present work, a series of suggestions are presented:

Abstract. Please specify the confidence interval (CI) or the p value from which results were considered significant i.e. p<0.05 / p<0.01.

Response: Respected reviewer, Specified p value and significance has been mentioned in the abstract of the revised manuscript

Introduction. “In this aspect, there is paucity in the literature regarding the effect of cordycepin on 81 the viability proliferation and migratory properties of dental pulp-derived mesenchymal 82 stem cells.” Please add a reference or references to support this statement.

Response: The comment of the esteemed reviewer is well taken. The information is derived from literature search and from ref [2]. It is the observation of the author. The reviewer may kindly consider.

Materials and methods. The methodology for the wound healing assay is not specified in the materials section, please include a paragraph to explain it.

Response: Respected reviewer, Included methodology for wound healing assay in the materials and methods section.

Discussion:

There are no references in the third paragraph from the discussion section. Please include references to support your statements.

Response: Respected reviewer, According to the suggestion of reviewers, the reference is now added.

The fifth paragraph in the discussion section only describes the results from the study. Authors should include a statement on the implications of such results and mention if there is previous evidence with similar or dissimilar results.Please specify the possible limitations of this study in further detail.Please include a paragraph on the possible clinical implications of this work, with references.It would also be interesting to include a paragraph on the future lines of research which could derive from this study (e.g. analysis of the influence of cordycepin on the osteo/odontogenic differentiation of dental pulp stem cells at an in vitro level), with references.

Response:Respected reviewer According to the suggestion of reviewers, the references, implications and limitations are now added.

Reviewer 4 Report

  1. Please include the clinical trial registry number.
  2. Please explain clearly in the abstract, specially the materials and method section.
  3. Poor sample size. n = 5
  4. Human third molar teeth were obtained from healthy subjects who were aged 14–25 years undergoing orthodontic tooth extraction with good oral hygiene (n = 5).........age 14 - 17 only tooth germs are there. Extraction for orthodontic purposes....please explain.
  5. Figures 1 A and 4 are of poor quality. Please increase magnification and replace.

Author Response

Reviewer:

  1. Please include the clinical trial registry number.

Response: Respected reviewer, This is an in vitro study and we have obtained appropriate ethical permissions required for the human tissue samples.The ethical approval number and details have been furnished in the manuscript

  1. Please explain clearly in the abstract, specially the materials and method section.

Respected reviewer, the details of the study are clearly mentioned in the abstract as suggested.

  1. Poor sample size. n = 5

Response: respected reviewer, we have done multiple experiments in duplicates for obtaining authentic results. It is to be reiterated that we cultured the stem cells upto many passages also for the study. Hence we could perform the experiment only on 5 teeth. Kindly consider our efforts and help us

  1. Human third molar teeth were obtained from healthy subjects who were aged 14–25 years undergoing orthodontic tooth extraction with good oral hygiene (n = 5).........age 14 - 17 only tooth germs are there. Extraction for orthodontic purposes....please explain.

Respected reviewer we humbly apologise for the typo error. We performed the experiments on premolar teeth extracted for orthodontic purpose and not on third molars. we have made the change in the manuscript

  1. Figures 1 A and 4 are of poor quality. Please increase magnification and replace.

Response: respected reviewer,  Replaced with high quality figures.

Round 2

Reviewer 1 Report

The paper has been improved by the authors and the aim  has been better focused. Basing on author's statements   5 microM Cordycepin stimulates osteoblast committed DPSC regenerative response and thus could be used in restorative dentistry. I suggest the publication of this paper

Reviewer 3 Report

Authors have answered to my concerns.